# Relationship Between Diaphragm Function and Sarcopenia Assessed by Ultrasound: A Cross-Sectional Study

**DOI:** 10.3390/diagnostics15010090

**Published:** 2025-01-03

**Authors:** Takahiro Shinohara, Toru Yamada, Shuji Ouchi, Suguru Mabuchi, Ryoichi Hanazawa, Kazuharu Nakagawa, Kanako Yoshimi, Tatsuya Mayama, Ayane Horike, Kenji Toyoshima, Yoshiaki Tamura, Atsushi Araki, Haruka Tohara, Akihiro Hirakawa, Takuma Kimura, Takeshi Ishida, Masayoshi Hashimoto

**Affiliations:** 1Department of General Medicine, Graduate School of Medical and Dental Sciences, Institute of Science Tokyo, Tokyo 113-8510, Japan; tshifmed@tmd.ac.jp (T.S.); ouchi.vasc@tmd.ac.jp (S.O.); mabuchi.vasc@tmd.ac.jp (S.M.); mikejpn007@gmail.com (M.H.); 2Department of Clinical Biostatistics, Graduate School of Medical and Dental Sciences, Institute of Science Tokyo, Tokyo 113-8510, Japan; r-hanazawa.crc@tmd.ac.jp (R.H.); a-hirakawa.crc@tmd.ac.jp (A.H.); 3Department of Dysphagia Rehabilitation, Department of Gerontology and Gerodontology, Graduate School of Medical and Dental Sciences, Institute of Science Tokyo, Tokyo 113-8510, Japan; k.nakagawa.swal@tmd.ac.jp (K.N.); k.yoshimi.gerd@tmd.ac.jp (K.Y.); gambatteiki@gmail.com (T.M.); ayanehorike55@gmail.com (A.H.); harukatohara@hotmail.com (H.T.); 4Department of Diabetes, Metabolism, and Endocrinology, Tokyo Metropolitan Institute for Geriatrics and Gerontology, Tokyo 173-0015, Japan; kenji_toyoshima@tmghig.jp (K.T.); tamurayo@tmghig.jp (Y.T.); aaraki@tmghig.jp (A.A.); 5Department of R&D Innovation for Home Care Medicine, Graduate School of Medical and Dental Sciences, Institute of Science Tokyo, Tokyo 152-8550, Japan; kimura.takuma@tmd.ac.jp; 6Department of Community Medicine (Ibaraki), Graduate School of Medical and Dental Sciences, Institute of Science Tokyo, Tokyo 152-8550, Japan; ishida.takeshi@tmd.ac.jp

**Keywords:** diaphragm, ultrasound, sarcopenia, point of care ultrasound

## Abstract

**Background/Objectives**: The diaphragm is important for respiration, but the effects of age-related muscle loss and sarcopenia on diaphragm function are unclear. We evaluated the associations of sarcopenia and skeletal muscle mass (SMM) with diaphragm function. **Methods**: This study was conducted at three Japanese hospitals from May 2023 to September 2024. The participants underwent bioelectrical impedance for SMM assessment, as well as pulmonary function tests. Diaphragm ultrasound was used to measure the thickness at functional residual capacity (FRC), thickening fraction (TF), and diaphragm excursion (DE) during deep breathing (DB), and their associations with sarcopenia and low skeletal muscle index (SMI) were analyzed. **Results**: Overall, 138 patients (mean age 78.0 years; sarcopenia, *n* = 35; non-sarcopenia, *n* = 103) were included. No statistically significant differences in thickness (FRC), TF and DE were observed between the sarcopenia group and the non-sarcopenia group. The low SMI group had significantly lower thickness (difference −0.22, 95% CI; −0.41, −0.29) and DE (difference −9.2, 95%CI; −14.0, −4.49) than the normal SMI group. Multivariable linear regression analyses adjusted for age, sex, and stature revealed no association between thickness (FRC) and sarcopenia (*p* = 0.98), but thickness (FRC) was negatively associated with low SMI (*p* = 0.034). DE during DB was negatively associated with sarcopenia (*p* = 0.024) and low SMI (*p* = 0.001). TF showed no associations. **Conclusions**: DE during DB was reduced in patients with sarcopenia and low SMI, and thickness (FRC) was reduced in those with low SMI without sarcopenia.

## 1. Introduction

Sarcopenia is defined as a progressive decline in skeletal muscle mass and strength, and is associated with falls, frailty, physical disability, hospitalization, and mortality [1,2]. The prevalence of sarcopenia among Japanese individuals aged ≥ 60 years is reported to be 8.5% in men and 8.0% in women based on the criteria of the Asian Working Group for Sarcopenia (AWGS) [3,4], making it an increasingly important issue in Japan’s aging society. Sarcopenia not only results in a decline in skeletal muscle mass and strength in the limbs, but it also leads to decreased respiratory function, a condition referred to as respiratory muscle sarcopenia or respiratory sarcopenia [5,6,7,8,9]. Previous studies have suggested that the respiratory muscles have a greater impact on physical function than overall body muscles [10], highlighting the importance of the respiratory muscles for sarcopenia evaluation.

The diaphragm is the primary respiratory muscle, accounting for approximately 75% of inspiratory effort [11,12]. As a skeletal muscle, the diaphragm may exhibit a decline in muscle mass and strength in patients with sarcopenia [13]. Diaphragm dysfunction is thought to lead to respiratory distress, decreased exercise capacity, and a lower quality of life (QoL) [14,15,16]. Diaphragm function can be assessed by ultrasound [17] using one of two measurement methods. The first method involves measuring the diaphragm thickness and the change in thickness due to respiration, known as the thickening fraction (TF). The second measures the movement of the dome of the diaphragm during respiration, known as diaphragm excursion (DE) [18]. Previous studies have not clearly established the association between diaphragm thickness and maximal inspiratory pressure (MIP) [19,20], but a correlation between DE and both MIP and maximal expiratory pressure (MEP) has been reported [20]. These findings suggest the potential for diaphragm ultrasound parameters to assess respiratory muscle strength and respiratory sarcopenia.

Currently, very few studies have evaluated diaphragm function using ultrasound in the context of sarcopenia. As indicated by previous research, individuals with sarcopenia have been reported to have a thinner diaphragm than those without sarcopenia [21]. Additionally, indicators of sarcopenia, such as appendicular skeletal muscle mass to body mass index (ASM/BMI), calf circumference, grip strength, and walking speed, have been found to correlate with thickness [22]. Regarding the relationship between DE and sarcopenia, a previous report indicated a correlation between a decrease in limb skeletal muscle mass and a reduction in DE [23]. However, these studies have several limitations. First, the diagnosis of sarcopenia was not based on international standards, and the participants were relatively young. Additionally, no studies have comprehensively evaluated thickness, TF, and DE in the same subjects. Consequently, it remains unclear which of thickness or DE is more closely associated with sarcopenia. Furthermore, no studies have investigated the correlation be-tween sarcopenia and TF.

In this study, we aim to investigate the relationship between thickness, TF, and DE measured by ultrasound, and the associations of these parameters with the presence of sarcopenia and decreased skeletal muscle mass in Japanese individuals aged ≥65 years, based on the sarcopenia criteria of the AWGS 2019.

## 2. Materials and Methods

### 2.1. Study Population and Setting

This study was conducted from May 2023 to September 2024 at the Swallowing Rehabilitation Outpatient Clinic of Tokyo Medical and Dental University (Tokyo, Japan), as well as the Frailty Clinic of Tokyo Metropolitan Institute for Geriatrics and Gerontology (Tokyo, Japan) and Hitachiomiya Saiseikai Hospital (Ibaraki, Japan), which provide specialized outpatient services for comprehensive geriatric functional assessment. In this study, data from outpatients aged ≥ 65 years were analyzed. Elderly individuals aged ≥ 65 years who were able to attend outpatient visits, could perform the instructed tasks, and provided consent to participate were included. Participants who could not provide informed consent, were receiving home oxygen therapy, and could not undergo bioimpedance analysis were excluded. This study did not include a priori sample size calculation because it was designed for exploratory analysis.

Medical history was obtained from all patients, including age, sex, underlying dis-eases, and medication history. Physical findings, such as stature and weight, were recorded. Grip strength was measured for both hands, and the higher value was used for the analysis. It is important to note that grip strength testing shows only a weak correlation with various measures of muscle strength and physical function [24]. Gait speed was measured by having the participants walk 6 m at a normal pace following markings on the ground. The speed was calculated based on the average value obtained from the distance excluding the first and last 1 m. Pulmonary function tests were conducted using a spirometer (AutoSpiro507 or AutoSpiroAS-610; Minato Medical Science Co., Ltd., Osaka, Japan, DISCOM-51; CHEST). Each parameter was measured twice, and the best result was used for the analysis. Skeletal muscle mass was measured in the standing position using bioelectrical impedance analysis (Inbody S10^®^ or Inbody770^®^; InBody Japan Inc., Tokyo, Japan). Sarcopenia was diagnosed using the AWGS 2019 criteria [25]. The definition of sarcopenia was based on the skeletal muscle index (SMI), which was calculated by dividing skeletal muscle mass by the square of stature. A diagnosis of sarcopenia was made for men with an SMI of <7.0 kg/m^2^ and for women with an SMI of <5.7 kg/m^2^, along with a gait speed of <1 m/s or a grip strength of <28 kg for men and <18 kg for women. The criteria for low skeletal muscle mass (hereafter referred to as the low SMI group) were also based on the diagnostic criteria of the AWGS 2019, defining low SMI as <7.0 kg/m^2^ for men and <5.7 kg/m^2^ for women, as measured by bioelectrical impedance analysis [25].

### 2.2. Diaphragm Ultrasound

The diaphragm ultrasound examination was performed by a physician or an ultrasound technician with sufficient training, following a protocol prepared with reference to previous studies [26,27]. All measurements were performed on the right side. The thickness was assessed in the seated position using B-mode ultrasound. A linear probe (7.5 MHz) was placed perpendicularly along the anterior to mid-axillary line of the 8th to 9th intercostal spaces, adjusted to avoid the ribs, and positioned to capture a partial image of the lung during inspiration. The thickness was measured at the end of quiet expiration (functional residual capacity [FRC]) and at maximum inspiration (total lung capacity [TLC]). During thickness measurement, verbal instructions were given as follows: “Take a deep breath in, and at the end of the inhalation, exhale and relax.” Deep inspiration was defined as TLC, and the end of exhalation was defined as FRC. The thickness was measured using electronic calipers as the distance between the central line of the pleura and peritoneum [26]. The TF was calculated using the following formula:Thickness (TLC) − thickness (FRC) ÷ thickness (FRC) × 100.

The DE was measured in the seated position using a sector probe (5 MHz) in M-mode. The probe was placed along the anterior to mid-axillary line of the 8th to 9th intercostal spaces and adjusted to avoid the ribs to visualize the diaphragm dome. The M-mode line was adjusted to be as perpendicular as possible to the diaphragm dome, and the difference in movement of the dome during deep breathing (DB) was measured [26]. The verbal instructions for measuring DE during DB were the same as those for thickness measurement: “Take a deep breath in, and at the end of the inhalation, exhale and relax”.

### 2.3. Statistical Analysis

Comparisons of patient characteristics, thickness (FRC), TF, and DE during DB be-tween the sarcopenia and non-sarcopenia groups were performed using the Student’s *t*-test for continuous variables and the chi-square test for categorical variables. The factors associated with each parameter (thickness [FRC], TF, and DE during DB) and sarcopenia were identified using univariate and multivariable linear regression analyses. The model included age, sex, and stature as covariates. The coefficient, 95% confidence interval, and *p* value were estimated for each covariate. Statistical significance was defined as a *p* value of <0.05. All data were analyzed using Stata, version 17.0 (StataCorp LLC, College Station, TX, USA).

## 3. Results

The mean participant age was 78.0 (standard deviation of 6.9) years, and 52.2% of the participants were female. There were no significant differences in sex, Brinkman Index, Charlson Comorbidity Index, Barthel Index, stature, forced expiratory volume in 1 s/forced vital capacity ratio, and peak expiratory flow rate between the sarcopenia group and the non-sarcopenia group. The sarcopenia group was older and had a lower Mini-Mental State Examination score, lower body mass, lower grip strength, and slower walking speed than the non-sarcopenia group. Additionally, the percent vital capacity, as an indicator of respiratory function, was lower in the sarcopenia group (Table 1).

### 3.1. Comparison Between the Sarcopenia and Non-Sarcopenia Groups

There were no statistically significant differences in diaphragm thickness (FRC), TF and DE during DB between the sarcopenia group and the non-sarcopenia group (Table 2).

In the simple regression analysis of the relationship between the presence or absence of sarcopenia and thickness (FRC), TF, and DE during DB, no association was observed between the presence of sarcopenia and thickness (FRC) (*p* = 0.85), TF (*p* = 0.44), or DE during DB (*p* = 0.09) [Table 3(a)].

In the multivariable linear regression analysis adjusted for sex, age, and stature, a statistically significant negative association was observed between DE during DB and sarcopenia (*p* = 0.024). No significant association was observed between sarcopenia and thickness (FRC) (*p* = 0.98) or between sarcopenia and TF (*p* = 0.28) [Table 3(b)].

### 3.2. Comparison Between the Low SMI and Normal SMI Groups

Diaphragm thickness (FRC) was significantly lower in the low SMI group than in the normal SMI group (difference −0.22, 95% CI; −0.41, −0.29). TF showed no statistically significant difference between the low SMI group and the normal SMI group (difference 9.8, 95% CI; −8.8, 28.4). DE during DB was significantly smaller in the low SMI group than in the normal SMI group (difference −9.2, 95%CI; −14.0, −4.49) (Table 4).

In the simple regression analysis of the relationships between SMI and thickness (FRC), TF, and DE (DB), low SMI was significantly negatively associated with thickness (FRC) (*p* = 0.024) and DE during DB (*p* < 0.001). No association was found with TF (*p* = 0.299) [Table 5(a)].

In the multiple regression analysis adjusted for sex, age, and stature, a statistically significant negative association was observed between low SMI and thickness (FRC) (*p* = 0.034) and between low SMI and DE during DB (*p* = 0.001). No significant association was found between TF and low SMI (*p* = 0.082) [Table 5(b)].

## 4. Discussion

This study investigated the associations of sarcopenia and low SMI with diaphragm thickness (FRC), TF, and DE during DB in elderly individuals aged ≥ 65 years. Compared with the non-sarcopenia group, the sarcopenia group showed no statistically significant differences. However, in the multivariable linear regression analysis adjusted for age, sex, and stature, DE during DB was significantly negatively associated with sarcopenia. In the subgroup analysis, the participants were divided into two groups (low and normal SMI) based on the sarcopenia diagnostic criteria [25]. The results showed that the low SMI group had a significantly lower diaphragm thickness (FRC) and a significantly smaller DE during DB than the normal SMI group. In the multiple regression analysis adjusted for age, sex, and stature, low SMI was significantly negatively associated with thickness (FRC) and DE during DB, while TF was not affected by the presence or absence of sarcopenia or the reduction in skeletal muscle mass.

### 4.1. Diaphragm Thickness

There was no significant difference in diaphragm thickness (FRC) between the sarcopenia group and the non-sarcopenia group. However, the low SMI group had a lower thickness (FRC) than the normal SMI group. In a previous study by Deniz et al., patients with sarcopenia, diagnosed according to the European Working Group on Sarcopenia in Older People diagnostic criteria, had a lower diaphragm thickness than those without sarcopenia [21]. Additionally, Lee et al. evaluated the relationship between sarcopenia indicators and respiratory muscle strength and thickness in healthy volunteers aged ≥ 65 years. It was reported that thickness was correlated with ASM/BMI, which is an indicator of appendicular skeletal muscle mass [22].

In the present study, there was no significant difference in thickness (FRC) between the sarcopenia group and the non-sarcopenia group. Meanwhile, a statistically significant negative association was observed between the reduction in skeletal muscle mass and thickness (FRC).

The diaphragm contains a similar proportion of slow-twitch fibers (type I fibers) and fast-twitch fibers (type II fibers), similar to the composition of other skeletal muscles. Therefore, in conditions that feature skeletal muscle wasting, such as sarcopenia, the muscle fibers of the diaphragm may be affected [28].

In animal studies, it has been reported that mice with sarcopenia show a decrease in fast twitch fibers (type IIx and IIb) in the diaphragm, along with atrophy and a reduction in cross-sectional area, with aging [29,30,31]. There have also been reports that the cross-sectional area of the fibers was larger in older mice than in younger mice, and that fat may infiltrate the diaphragm and increase its thickness with age [32,33,34].

In this study, we compared older adults with and without sarcopenia. Therefore, various factors, such as muscle atrophy due to sarcopenia and pseudo-hypertrophy due to aging, may have influenced the outcomes, possibly explaining why the thickness was not lower in the sarcopenia group.

The diagnostic criteria for sarcopenia not only include a decrease in muscle mass, but also a decline in muscle strength. Therefore, although the diaphragm was thinner in the low SMI group, it may not have been thinner in the sarcopenia group owing to the influence of the muscle strength component of the classification.

### 4.2. Diaphragm Thickening Fraction

In this study, there was no significant difference in TF between the sarcopenia and non-sarcopenia groups, or between the low SMI and normal SMI groups. To date, no studies have investigated the association between the reduction in skeletal muscle mass based on the sarcopenia diagnostic criteria and TF in older adults. A previous study involving healthy older adults indicated that TF was not affected by factors such as sex, stature, and BMI [35]. The results of the present study also indicated that TF was not affected by sarcopenia or reduced skeletal muscle mass. TF may not be suitable for the diagnosis of sarcopenia or reduced skeletal muscle mass. However, as it is less influenced by factors such as age, height, and sex, it has the potential to be applicable as a diagnostic criterion for diaphragmatic paralysis across a wide range of patients [26].

### 4.3. Diaphragm Dome Excursion

In this study, DE during DB was negatively associated with sarcopenia and low SMI. Zeng et al. reported that reduced skeletal muscle mass was associated with decreased DE in older adults who were found to have pulmonary nodules during community health screening [23]. Therefore, the findings of the present study support the results of previous research.

An animal study reported that mice with sarcopenia have weaker respiratory muscle strength [29], and that aged mice with sarcopenia exhibit a reduction in maximal specific contractile force [30]. In humans, a positive correlation between maximum inspiratory pressure and DE during DB has also been reported. The mechanism is thought to involve a larger DE resulting in a greater change in thoracic cavity volume, which increases the negative pressure exerted on the lungs, ultimately leading to an increase in MIP [20]. These previous studies have also suggested an association between sarcopenia or reduced skeletal muscle mass and DE during DB. The diaphragm consists of fatigue-resistant slow-twitch fibers and fast-twitch fibers, with the latter being more prone to fatigue but capable of generating rapid, powerful contractions. Sarcopenia is associated with a reduction in fast twitch fibers only [30]. The ventilatory action of the diaphragm during normal breathing is maintained in old age. However, non-ventilatory actions that require greater force, such as deep breathing, are thought to be impaired in those with sarcopenia [36]. This may reflect the age-related reduction in fast twitch fibers and could manifest as a de-crease in DE during DB. In addition, multivariable linear regression analysis revealed that DE was statistically significantly associated with stature. Previous studies have reported that stature does not influence DE in younger individuals [20], suggesting that this association may be specific to older adults. One possible factor is vertebral deformity, such as kyphosis, caused by decreased bone density, which may affect stature and consequently influence DE. However, vertebral deformities were not measured in this study, and further investigations are necessary to clarify this relationship.

In summary, DE during DB decreases with sarcopenia and reduced skeletal muscle mass. This suggests that DE during DB may have the potential to detect diaphragmatic changes due to sarcopenia and skeletal muscle loss more effectively than diaphragm thickness, which is influenced by factors such as muscle atrophy and fat infiltration.

In recent years, the concept of respiratory sarcopenia, which refers to the decline in respiratory muscle function, has been proposed [37]. It has been reported that a decline in respiratory muscle strength affects physical function, leading to reduced exercise capacity, decreased walking ability, and reduced QoL [38]. Given this background, preventing the decline in respiratory muscle strength may reduce the risk of complications and mortality, helping to mitigate functional decline in older adults. Consequently, respiratory sarcopenia is becoming an increasingly important topic in the aging society [39]. The diaphragm is the primary muscle responsible for respiratory movement, and it is an essential factor in the assessment, diagnosis, and treatment of respiratory sarcopenia [11].

Diaphragm ultrasound is a simple and non-invasive examination, and it has shown good reproducibility with appropriate training [40]. Although diaphragm ultrasound has been shown to be useful in the diagnosis of respiratory sarcopenia [9], specific methods and standards have not been established. Our study clarified which aspects of diaphragm function are associated with sarcopenia and skeletal muscle reduction. A future challenge is to establish specific cutoff values for respiratory sarcopenia. The present study also suggested that the diagnostic criteria for systemic sarcopenia are heavily influenced by the skeletal muscles of the limbs, indicating limitations in assessing the impact on diaphragmatic function based solely on the presence or absence of systemic sarcopenia.

### 4.4. Limitations

This study has several limitations. First, this study focused on older adults aged ≥65 years, so it is unclear whether the findings are applicable to other age groups. Structural changes in diaphragm fibers have been reported with aging [41], and skeletal muscle loss in younger individuals may yield different results, warranting further investigation.

Second, although the ultrasound examinations were conducted by pre-trained examiners, inter-observer variability was not evaluated, and differences in measurement accuracy between examiners have not been verified. Third, the study included patients who visited medical institutions, and we did not examine the possibility that underlying conditions may have influenced the results. The Charlson Comorbidity Index was low, and the Barthel Index was very high, suggesting that the patients had well-maintained physical function and low severity of illness. This is similar to the typical outpatient population, so it is likely that some generalization is possible.

## 5. Conclusions

This study revealed that DE during DB, as assessed by diaphragm ultrasound, has a negative association with sarcopenia and low SMI. Additionally, diaphragm thickness (FRC) was not related to sarcopenia, but it was associated with low SMI. TF was not influenced by these factors and may potentially serve as a consistent reference across a wide range of patient populations. In the future, it will be necessary to establish reference values for DE during DB and thickness (FRC) in relation to sarcopenia and respiratory sarcopenia.

## Figures and Tables

**Table 1 diagnostics-15-00090-t001:** Participants’ characteristics.

	Non-Sarcopenia (*n* = 103)	Sarcopenia (*n* = 35)	*p* Value
Age (years), mean (SD)	76.6 (6.81)	81.9 (5.36)	<0.001
Female, *n* (%)	56 (54.4)	16 (45.7)	0.49
Never smoker, *n* (%)	59 (57.2)	22 (62.9)	
Current smoker, *n* (%)	7 (6.8)	2 (5.7)	
Past smoker, *n* (%)	36 (35.0)	11 (31.4)	
No answer, *n* (%)	1 (1.0)	0 (0.0)	
Brinkman Index, mean (SD)	820.4 (721)	732 (436)	0.689
Charlson Comorbidity Index, mean (SD)	1.37 (1.5)	1.66 (1.3)	0.316
Barthel Index, mean (SD)	98.5 (4.2)	97.4 (3.9)	0.205
MMSE, mean (SD)	27.9 (2.5)	25.4 (5.0)	<0.001
Stature (cm), mean (SD)	156.7 (9.4)	154.3 (12.1)	0.225
Body mass (kg), mean (SD)	58.6 (10.9)	51.8 (9.4)	0.001
Grip strength (kg), mean (SD)	26.3 (7.1)	20.8 (7.6)	<0.001
Gait speed (m/s), mean (SD)	1.2 (0.3)	0.97 (0.3)	<0.001
SMI (kg/m^2^), mean (SD)	6.7 (1.0)	6 (1)	<0.001
Pulmonary function test			
%VC (%), mean (SD)	95 (17.3)	85.5 (16.3)	0.007
FEV_1_/FVC (%), mean (SD)	78.7 (7.8)	80.6 (10.6)	0.273
Peak expiratory flow rate (L/min), mean (SD)	4.9 (1.6)	4.8 (2)	0.211

MMSE: Mini-Mental State Examination; SMI: skeletal muscle mass index; %VC: percent vital capacity; FEV_1_/FVC: forced expiratory volume in 1 s/forced vital capacity ratio; SD: standard deviation.

**Table 2 diagnostics-15-00090-t002:** Comparison of Diaphragm thickness, Thickening fraction, and Diaphragm excursion between non-sarcopenia and sarcopenia groups.

Parameter	Total	Non-Sarcopenia	Sarcopenia	Difference (95% CI)
Diaphragm thickness at FRC (mm), *n* = 136	2.04 (0.56)	2.03 (0.56)	2.05 (0.6)	0.02 (−0.2, 0.24)
Thickening fraction (%), *n* = 136	89.3 (54.1)	87.2 (50)	95.4 (65.0)	8.2 (−12.8, 29.2)
Diaphragm excursion (mm), *n* = 128	44 (14.2)	45.3 (14.7)	40.4 (12.3)	−4.9 (−10.5, 0.69)

Data are presented as the mean (SD). FRC: functional residual capacity; SD: standard deviation. CI: confidence interval. Sarcopenia includes participants who met the diagnostic criteria for sarcopenia of the AWGS 2019. Non-sarcopenia includes participants who did not meet the diagnostic criteria of the AWGS 2019. Difference was calculated as the mean of the sarcopenia group minus the non-sarcopenia group.

**Table 3 diagnostics-15-00090-t003:** Results of the simple and multiple regression analyses for sarcopenia and each right hemidiaphragm measurement.

(a). Simple regression analysis for right hemidiaphragm thickness, thickening fraction, and excursion						
Independent Variables Parameter	Diaphragm Thickness at FRC (mm), *n* = 136 Coefficient [95% CI] *p* Value		Thickening Fraction (%), *n* = 136 Coefficient [95% CI] *p* Value		Diaphragm Excursion (mm), *n* = 128 Coefficient [95% CI] *p* Value	
Sarcopenia (reference: non-sarcopenia)	0.02 [−0.2, 0.24]	0.85	8.2 [−12.8, 29.2]	0.44	−4.9 [−10.5, 0.7]	0.09
(b). Multiple regression analysis for each right hemidiaphragm measurement with adjustment for independent variables.						
Independent Variables Parameter	Diaphragm Thickness at FRC (mm), *n* = 136 Coefficient [95% CI] *p* Value		Thickening Fraction (%), *n* = 136 Coefficient [95% CI] *p* Value		Diaphragm Excursion (mm), *n* = 128 Coefficient [95% CI] *p* Value	
Sarcopenia (reference: non-sarcopenia)	0.004 [−0.23, 0.24]	0.98	12.1 [−10.2, 34.4]	0.28	−6.3 [−11.7, −0.85]	0.024 *
Age	0.007 [−0.008, 0.23]	0.35	−0.25 [−1.7, 1.2]	0.73	0.5 [0.16, 0.9]	0.005 *
Female sex (reference: male)	0.02 [−0.26, 0.3]	0.89	2.5 [−24, 29]	0.85	−1.2 [−7.7, 5.3]	0.7
Stature	0.009 [−0.005, 0.02]	0.224	1.1 [−0.23, 2.5]	0.1	0.56 [0.23, 0.9]	0.001 *

CI: confidence interval; FRC: functional residual capacity. Sarcopenia includes participants who met the diagnostic criteria for sarcopenia of the AWGS 2019. *: *p* < 0.05.

**Table 4 diagnostics-15-00090-t004:** Comparison of Diaphragm thickness, Thickening fraction, and Diaphragm excursion between Normal SMI and Low SMI groups.

Parameter	Total	Normal SMI	Low SMI	Difference (95% CI)
Diaphragm thickness at FRC (mm), *n* = 136	2.04 (0.56)	2.13 (0.6)	1.91 (0.54)	−0.22 (−0.41, −0.29)
Thickening fraction (%), *n* = 136	89.3 (54.1)	85 (49)	95 (60)	9.8 (−8.8, 28.4)
Diaphragm excursion (mm), *n* = 128	44.0 (14.2)	48.2 (14.7)	38.9 (11.8)	−9.2 (−14.0, −4.49)

Data are presented as the mean (SD). FRC: functional residual capacity; SD: standard deviation. CI: confidence interval. Low SMI includes participants who met the SMI cutoff defined in the diagnostic criteria for sarcopenia of the AWGS 2019. Normal SMI includes participants who did not meet the SMI cutoff defined in the diagnostic criteria for sarcopenia of the AWGS 2019. Difference was calculated as the mean of the sarcopenia group minus the non-sarcopenia group.

**Table 5 diagnostics-15-00090-t005:** Results of the simple and multiple regression analyses for skeletal muscle mass and each right hemidiaphragm measurement.

(a). Simple regression analysis for right hemidiaphragm thickness, thickening fraction, and excursion						
Independent Variables Parameter	Diaphragm Thickness at FRC (mm), *n* = 136 Coefficient [95% CI] *p* Value		Thickening Fraction (%), *n* = 136 Coefficient [95% CI] *p* Value		Diaphragm Excursion (mm), *n* = 128 Coefficient [95% CI] *p* Value	
Low SMI (reference: normal SMI)	−0.22 [−0.41, −0.29]	0.024 *	9.8 [−8.8, 28.4]	0.299	−9.2 [−14.0, −4.4]	<0.001 *
(b). Multiple regression analysis for each right hemidiaphragm measurement with adjustment for independent variables						
Independent Variables Parameter	Diaphragm Thickness at FRC (mm), *n* = 136 Coefficient [95% CI] *p* Value		Thickening Fraction (%), *n* = 136 Coefficient [95% CI] *p* Value		Diaphragm Excursion (mm), *n* = 128 Coefficient [95% CI] *p* Value	
Low SMI (reference: normal SMI)	−0.22 [−0.42, −0.017]	0.034 *	17.2 [−2.2, 36.5]	0.082	−8 [−12.7, −3.4]	0.001 *
Age	0.01 [−0.005, 0.24]	0.206	−0.2 [−1.6, 1.2]	0.789	0.5 [0.13, 0.82]	0.007 *
Female sex (reference: male)	−0.01 [−0.28, 0.26]	0.936	2.9 [−23.2, 29]	0.826	−1.5 [−7.8, 4.8]	0.639
Stature	0.005 [−0.09, 0.02]	0.473	1.3 [−0.05, 2.7]	0.058	0.5 [0.14, 0.81]	0.006 *

CI: confidence interval; FRC: functional residual capacity. Low SMI includes participants who met the SMI cutoff defined in the diagnostic criteria for sarcopenia of the AWGS 2019. *: *p* < 0.05.

## Data Availability

The original contributions presented in this study are included in the article. Further inquiries can be directed to the corresponding author(s).

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
