# Peer review of "Relationship Between Diaphragm Function and Sarcopenia Assessed by Ultrasound: A Cross-Sectional Study"

_diagnostics, 2025, doi:10.3390/diagnostics15010090_

Round 1
Reviewer 1 Report
Comments and Suggestions for Authors
This article is novel and very informative
Author Response
Dear Reviewer
Thank you very much for taking the time to read our manuscript. We sincerely appreciate your positive evaluation and thoughtful comments. We are truly grateful for your time and effort.
We look forward to your continued support and guidance. Thank you.
Reviewer 2 Report
Comments and Suggestions for Authors
Thank you for selecting me as a reviewer.
This study investigated the impact of sarcopenia and skeletal muscle mass (SMM) on diaphragm function in older adults. Conducted across three hospitals in Japan, the study included 148 participants (mean age: 78.1 years), with 35 diagnosed with sarcopenia. The assessments performed were bioelectrical impedance analysis for SMM, pulmonary function tests, and diaphragm ultrasound, measuring thickness at functional residual capacity (FRC), thickening fraction (TF), and diaphragm excursion (DE) during deep breathing (DB).
Key findings include:
The authors concluded that DE during DB was impaired in individuals with sarcopenia and those with low skeletal mass index (SMI). At the same time, the reduction in FRC was primarily associated with low SMI. The thickening fraction was not found to be related to sarcopenia or SMM.
This study provides valuable insights into the relationship between muscle health and respiratory mechanics, particularly among aging populations. The findings emphasize the potential usefulness of diaphragm excursion as a functional marker for sarcopenia and low skeletal muscle mass. However, several methodological questions arose while reading the manuscript:
- The authors collected data from three hospitals, Hance, and the random effect of the hospital should be accounted for in the model. A random effect is significant when there is evidence of clustering in the data (e.g., measurements within the same hospital being more similar than between hospitals). Please provide evidence.
- Did the authors check whether the data fits the parametric statistical tests? Please check and report the assumptions of applying those statistical models.
- Finally, I suggest building a regression model in blocks. In the first block, add covariates (age, sex..), then explore the variation related to the sarcopenia factors. This will allow insight into variation change (R2 change) with additional variables. This would allow 1) Assessing Incremental Contribution of Variables, 2) Insight into R² Change, 3) Detecting Multicollinearity and Confounding, and 4) Improved Model Interpretation.
- If hospitals should be included in the model and the assumptions for the regression model are not met, the linear mixed-model effect is one of the reasonable models.
Author Response
Comment 1
The authors collected data from three hospitals, Hance, and the random effect of the hospital should be accounted for in the model. A random effect is significant when there is evidence of clustering in the data (e.g., measurements within the same hospital being more similar than between hospitals). Please provide evidence.
Response 1
Thank you for your feedback. I have consulted with the statistician who is a co-author of this study.
As you correctly pointed out, this study was conducted across three institutions, and the possibility of random effects cannot be ruled out. It should be noted that the examiners received prior instruction from the researchers on the examination procedures, and measures were taken to ensure the quality of the examinations. Additionally, regarding the patient background, the study included elderly patients capable of outpatient visits, making it unlikely for there to be significant variability in disease severity. Taking the above into consideration, we verified the random effects as a precaution.
Mixed-effect model:
Sarcopenia and DT, TF, DE (p = 0.974, 0.261, 0.012, respectively)
Low skeletal muscle mass and DT, TF, DE (p = 0.029, 0.067, <0.001, respectively)
Random-effect model:
Sarcopenia and DT, TF, DE (p = 0.975, 0.282, 0.022, respectively)
Low skeletal muscle mass and DT, TF, DE (p = 0.032, 0.080, <0.001, respectively)
Fixed-effect model:
Sarcopenia and DT, TF, DE (p = 0.986, 0.274, 0.014, respectively)
Low skeletal muscle mass and DT, TF, DE (p = 0.032, 0.072, <0.001, respectively)
Based on the above results, the statistical outcomes for the presence or absence of sarcopenia and low skeletal muscle mass in relation to the dependent variables were consistent.
As there were no differences in the results across the models, we have decided not to revise the content of this paper. Thank you for your valuable feedback.
Comment 2
Did the authors check whether the data fits the parametric statistical tests? Please check and report the assumptions of applying those statistical models.
Response 2
Thank you for your feedback. Although it was not mentioned in the main text, we confirmed the normality of the dependent variables—thickness at FRC, thickening fraction, and dome excursion—using histograms and Q-Q plots. Based on this confirmation, we determined that parametric tests were appropriate and conducted the analysis accordingly. Based on the above, we have not made any revisions to the content of this paper. Thank you for your valuable comments.
Comment 3
Finally, I suggest building a regression model in blocks. In the first block, add covariates (age, sex..), then explore the variation related to the sarcopenia factors. This will allow insight into variation change (R2 change) with additional variables. This would allow 1) Assessing Incremental Contribution of Variables, 2) Insight into R² Change, 3) Detecting Multicollinearity and Confounding, and 4) Improved Model Interpretation.
If hospitals should be included in the model and the assumptions for the regression model are not met, the linear mixed-model effect is one of the reasonable models.
Response 3
Thank you for your feedback. I have consulted with the statistician who is a co-author of this study.
As suggested, we have conducted a reanalysis accordingly.
1 Multivariate analysis excluding the variable for sarcopenia
・For thickness at FRC, no significant associations were observed with age, sex, or height (p-values: 0.202, 0.919, 0.270, respectively). The R² value was 0.0225.
・For thickening fraction, no significant associations were observed with age, sex, or height (p-values: 0.967, 0.982, 0.133, respectively). The R² value was 0.0349.
・For dome excursion, significant associations were observed with age and height (p-values: 0.021 and 0.001, respectively). No association was found with sex (p-value: 0.956). The R² value was 0.1719.
2 Results after adding the presence or absence of sarcopenia as a variable to the above models
The R² values improved in all models compared to those excluding sarcopenia:
・Thickness at FRC model: R² = 0.226
・Thickening fraction model: R² = 0.448
・Dome excursion model: R² = 0.2169
3 Multicollinearity Assessment using VIF
We tested for multicollinearity, and no variables showed a VIF exceeding 10 in any of the models:
・Thickness at FRC model: VIF values were height = 2.2, sex = 2.1, age = 1.23, and presence of sarcopenia = 1.15.
・Thickening fraction model: VIF values were height = 2.2, sex = 2.1, age = 1.23, and presence of sarcopenia = 1.15.
・Dome excursion model: VIF values were height = 2.17, sex = 2.08, age = 1.21, and presence of sarcopenia = 1.14.
4 Comparison between models with and without the presence of sarcopenia
When comparing the models excluding and including the presence of sarcopenia, the associations with each variable remained unchanged. The R² values were higher in the models that included sarcopenia, and the impact of multicollinearity was not significant. Therefore, we concluded that the models including the presence of sarcopenia are superior.
Based on the above, we determined that the models including the presence of sarcopenia are superior and that multicollinearity is not an issue. Therefore, we have decided to retain the models presented in the paper, and no revisions were made to the Table. Thank you for your valuable feedback.
Reviewer 3 Report
Comments and Suggestions for Authors
Dear Editor,
Thank you for the opportunity to review this study. The Authors reported the evaluation of “respiratory sarcopenia” in a cohort of patients older than 65 years.
The introduction is focused on describe the diagnostic problem of this new clinical condition and the materials and methods are clear described but I think there is a mistake. In the AWGS criteria, ASM was considered and cut-off reported not the SMI as describe in the methods. This aspect needs to be clarified because also the results might be changed.
The result section reported different variables considered and the statistic on simple, linear and multiple correlation. I think that a lack of this section is the evaluation of functional measure such as hand grip or gait speed that, together with body weight and pulmonary functional test, were significant different between the two groups.
In the same manner discussion needs to be reconsider. In fact, the Authors not evaluate the effects of sarcopenia but only the low of SMI (or ASM). Same consideration (e.g. page 9, line 305/309 and page 10 line 320/323) probably are not correct.
Author Response
Comment 1
The introduction is focused on describe the diagnostic problem of this new clinical condition and the materials and methods are clear described but I think there is a mistake. In the AWGS criteria, ASM was considered and cut-off reported not the SMI as describe in the methods. This aspect needs to be clarified because also the results might be changed.
In the same manner discussion needs to be reconsider. In fact, the Authors not evaluate the effects of sarcopenia but only the low of SMI (or ASM). Same consideration (e.g. page 9, line 305/309 and page 10 line 320/323) probably are not correct.
Response 1
Thank you for your feedback. As you correctly pointed out, the AWGS 2019 sarcopenia diagnostic criteria include a reduction in appendicular skeletal muscle mass (ASM) as a mandatory component *. For the assessment of ASM, Dual-energy X-ray absorptiometry (DXA) or Bioelectrical Impedance Analysis (BIA) is recommended, and the cutoff values are based on the Skeletal Muscle Index (SMI), calculated by dividing ASM by the square of height.
In this study, we adopted SMI values measured using BIA for the evaluation of sarcopenia, which aligns with the AWGS 2019 consensus. Therefore, we have not made any revisions to the content of the manuscript. Thank you for your valuable comments.
* Chen LK, Woo J, Assantachai P, Auyeung TW, Chou MY, Iijima K, et al. Asian Working Group for Sarcopenia: 2019 Consensus Update on Sarcopenia Diagnosis and Treatment. J Am Med Dir Assoc. 2020;21(3):300-7.e2.
Comment 2
The result section reported different variables considered and the statistic on simple, linear and multiple correlation. I think that a lack of this section is the evaluation of functional measure such as hand grip or gait speed that, together with body weight and pulmonary functional test, were significant different between the two groups.
Response 2
Thank you for your feedback. As you correctly pointed out, there were differences in handgrip strength and walking speed between the two groups. However, since these variables are included in the diagnostic criteria for sarcopenia, we did not include them as explanatory variables in the multivariate analysis to account for multicollinearity. Thank you for your valuable comments.
Reviewer 4 Report
Comments and Suggestions for Authors
The manuscript was designed to investigate the relationship between the thickness (DT), thickness fraction (TF), and excursion (DE) of the diaphragm muscle assessed via ultrasound and to identify associations between these parameters and sarcopenia in older adults. The study uses a rationale that sarcopenia also impacts respiratory muscles. It focuses on the diaphragm muscle as it is the main muscle involved in the respiratory cycle.
Although the relationship between muscle sarcopenia and respiratory muscle sarcopenia has been established, reduced physical activity may be the main factor mediating respiratory sarcopenia rather than sarcopenia as a general phenomenon. In addition, there is a general decrease in muscle function as age progresses.
Conceptually, it is mentioned that DT, TF, and DE are indicators of muscle function (line 55), but they are more related to morphological parameters. Therefore, the rationale of the introduction must be adjusted to accommodate such conceptual issues. I would expect that other respiratory parameters were included. They were measured but not mentioned in the introduction.
In addition, after reading some arguments, I wonder what the present study contributes beyond what has been already reported – see your references #19, #20, and #21 (lines 62, 65, and 67). The authors argue that such measurements derive from the same subjects, but a stronger rationale is required to sustain that the relationship between DT or DE and sarcopenia is unclear.
Methodologically, I have to disagree that 35 participants are outstanding and put the current study far ahead of others. Actually, estimating the sample size (not presented) is mandatory. This is especially concerning as you have split the sample in two (low and normal SMI). This is a novelty, only found in the results and discussion sections, as nothing is mentioned in the statistical analysis description.
Handgrip strength (HGS) is an inaccurate measure. Please, refer to the work conducted by Rodacki and colleagues. You will notice that HGS is poorly correlated with several strength and functional measures. It is hard to base your findings on such parameters.
The 4m walking test seems awkward as the speed was actually measured through only 2m, which may be insufficient to characterize walking speed properly.
Physical activity level must be considered as an important intervening variable.
The accuracy and reliability of the ultrasound measurements are crucial for the present work. A detailed analysis of such procedures is demanded.
Replace height with stature and body weight with body mass.
The statistics are equivocal, as nothing is mentioned to support the use of parametric tests. In addition, the description of the multivariate analysis is poor. No caution was exercised to include some variables. For instance, the sarcopenia definition includes all your covariates, and a large colinearity may exist. Furthermore, repeating t-tests is not recommended as it increases the odds of errors. Effect sizes and confidence intervals are required. There is also a bias in your data since the sarcopenia group was older. Thus, it is not known whether such changes are age-related or not. In addition, mixing women and men may have also inserted a bias as the small number of women (in a reduced group; 3 times smaller) may have implications.
The results are misinterpreted as non-significant differences are treated as “a trend”. This is inappropriate and misleading and unnaceptable.
In your multivariate analysis, colinearity must be reemphasized as an important issue. How was it controlled? In addition, table 3.2 does not present all the parameters necessary to characterize the analysis properly.
There are two tables 3.
Finally, most of the discussion is unfocused, as it is based on your inaccurate statistics. The discussion requires large revision as it replicates your results without gaining deeper insights.
Author Response
Comment 1
it is mentioned that DT, TF, and DE are indicators of muscle function (line 55), but they are more related to morphological parameters. Therefore, the rationale of the introduction must be adjusted to accommodate such conceptual issues. I would expect that other respiratory parameters were included. They were measured but not mentioned in the introduction.
Response 1
Thank you for your feedback. As you pointed out, there have been several reports on the relationship between diaphragm ultrasound indices and respiratory muscles*1,2. To improve clarity, we have added this information to the Introduction.
*1 Summerhill EM, Angov N, Garber C, McCool FD. Respiratory muscle strength in the physically active elderly. Lung. 2007;185(6):315-20.
*2 Yamada T, Minami T, Yoshino S, Emoto K, Mabuchi S, Hanazawa R, et al. Relationship Between Diaphragm Thickness, Thickening Fraction, Dome Excursion, and Respiratory Pressures in Healthy Subjects: An Ultrasound Study. Lung. 2024;202(2):171-8.
Before Revision (Lines 56-59)
One involves measuring the diaphragm thickness (DT) and the change in thickness due to respiration, known as the thickening fraction (TF). The other measures the movement of the dome of the diaphragm during respiration, known as diaphragm excursion (DE) [18].
After Revision (Lines 57-65)
One involves measuring the diaphragm thickness (DT) and the change in thickness due to respiration, known as the thickening fraction (TF). The other measures the movement of the dome of the diaphragm during respiration, known as diaphragm excursion (DE) [18] . Previous studies have reported a positive association be-tween diaphragm thickness (DT) and maximal inspiratory pressure (MIP) [19], as well as a correlation between diaphragm excursion (DE) and both MIP and maximal expiratory pressure (MEP) [20]. These findings suggest the potential for diaphragm ultrasound parameters to assess respiratory muscle strength and respiratory sarcopenia. 
Comment 2
after reading some arguments, I wonder what the present study contributes beyond what has been already reported – see your references #19, #20, and #21 (lines 62, 65, and 67). The authors argue that such measurements derive from the same subjects, but a stronger rationale is required to sustain that the relationship between DT or DE and sarcopenia is unclear.
Response 2
Thank you for your feedback. As you correctly pointed out, several previous studies have investigated the relationship between diaphragm ultrasound and sarcopenia. However, among these studies, only Deniz et al. evaluated diaphragm thickness based on international criteria for sarcopenia diagnosis (EWGSOP and AWGS criteria) *1, and no studies have assessed thickening fraction (TF) or diaphragm excursion (DE). Furthermore, no studies have comprehensively evaluated thickness, thickening fraction (TF), and diaphragm excursion (DE) together, nor have they examined which parameters are most suitable for clinical application. This has also been highlighted in review articles on the relationship between sarcopenia and diaphragm ultrasound *2. Based on the above, we believe that this study offers novelty compared to previous research. We have revised the description regarding the limitations of prior studies accordingly. Thank you for your valuable feedback.
*1 Deniz O, Coteli S, Karatoprak NB, Pence MC, Varan HD, Kizilarslanoglu MC, et al. Diaphragmatic muscle thickness in older people with and without sarcopenia. Aging Clin Exp Res. 2021;33(3):573-80.
*2 Siniscalchi C, Nouvenne A, Cerundolo N, Meschi T, Ticinesi A, On Behalf Of The Parma Post-Graduate Specialization School In Emergency-Urgency Medicine Interest Group On Thoracic U. Diaphragm Ultrasound in Different Clinical Scenarios: A Review with a Focus on Older Patients. Geriatrics (Basel). 2024;9(3).
Before Revision (Lines 67-70)
However, all of these studies were small-scale studies, and no studies have evaluated DT, TF, and DE in the same subjects. Therefore, it remains unclear which of DT or DE is more closely related to sarcopenia. Moreover, no studies have evaluated the correlation of sarcopenia with TF.
After Revision (Lines 73-79)
However, these studies have several limitations. First, the diagnosis of sarcopenia was not based on international standards, and the participants were relatively young. Additionally, no studies have comprehensively evaluated DT (diaphragmatic thickness), TF (thick-ness fraction), and DE (diaphragmatic echogenicity) in the same subjects. Consequently, it remains unclear which of DT or DE is more closely associated with sarcopenia. Further-more, no studies have investigated the correlation be-tween sarcopenia and TF.
Comment 3
Method
I have to disagree that 35 participants are outstanding and put the current study far ahead of others. Actually, estimating the sample size (not presented) is mandatory. This is especially concerning as you have split the sample in two (low and normal SMI). This is a novelty, only found in the results and discussion sections, as nothing is mentioned in the statistical analysis description.
Response 3
Thank you for your feedback. As you correctly pointed out, we did not perform a priori sample size calculation for this study. The reason is that this study was exploratory in nature. We have added a statement in the Methods section to clarify that a sample size calculation was not conducted.
Before Revision (Lines 84-86)
Participants who could not provide informed consent, were receiving home oxygen therapy, and could not undergo bioimpedance analysis were excluded.
After Revision (Lines 93-96)
Participants who could not provide informed con-sent, were receiving home oxygen therapy, and could not undergo bioimpedance analysis were excluded. This study did not include a priori sample size calculation because it was designed for exploratory analysis.
Comment 4
Method
Handgrip strength (HGS) is an inaccurate measure. Please, refer to the work conducted by Rodacki and colleagues. You will notice that HGS is poorly correlated with several strength and functional measures. It is hard to base your findings on such parameters.
Response 4
Thank you for your feedback. In this study, handgrip strength was measured for the diagnosis of sarcopenia based on the AWGS 2019 consensus *1 . However, as you pointed out, handgrip strength has been reported to show weak correlations with various muscle strengths and physical functions. We have added this point accordingly *2.
*1 Chen LK, Woo J, Assantachai P, Auyeung TW, Chou MY, Iijima K, et al. Asian Working Group for Sarcopenia: 2019 Consensus Update on Sarcopenia Diagnosis and Treatment. J Am Med Dir Assoc. 2020;21(3):300-7.e2.
*2 Rodacki ALF, Boneti Moreira N, Pitta A, Wolf R, Melo Filho J, Rodacki CLN, et al. Is Handgrip Strength a Useful Measure to Evaluate Lower Limb Strength and Functional Performance in Older Women? Clin Interv Aging. 2020;15:1045-56.
Before Revision (Lines 89-90)
Grip strength was measured for both hands, and the higher value was used for the analysis.
After Revision (Lines 99-101)
Grip strength was measured for both hands, and the higher value was used for the analysis. It is important to note that grip strength testing shows only a weak correlation with various measures of muscle strength and physical function [24].
Comment 5
Method
The 4m walking test seems awkward as the speed was actually measured through only 2m, which may be insufficient to characterize walking speed properly.
Response 5
Thank you for your feedback. As you correctly pointed out, the average speed over 2 meters is insufficient. Therefore, in this study, we adopted a walking distance of 6 meters to measure walking speed. To improve clarity, we have revised the relevant description in the manuscript.
Before Revision (Lines 90-93)
Gait speed was measured by having the participants walk a distance of 6 meters at a normal pace following markings on the ground. The average speed was calculated based on the time taken to cover the central 4 meters, excluding the first and last 1 meter.
After Revision (Lines 101-104)
Gait speed was measured by having the participants walk a distance of 6 meters at a normal pace following markings on the ground. The speed was calculated based on the average value obtained from the distance excluding the first and last 1 meter.
Comment 6
Method
Physical activity level must be considered as an important intervening variable.
Response 6
Thank you for your feedback. In this study, the Barthel Index, an indicator of physical activity levels, was very high in both the sarcopenia group (97.4 ± 3.9) and the non-sarcopenia group (98.5 ± 4.2), suggesting that the participants maintained their ADL. Additionally, since there was no statistically significant difference between the two groups (p = 0.205), we did not include it as an adjustment factor.
As a precaution, we conducted an analysis including the Barthel Index as a variable. However, the results regarding the association between sarcopenia and each outcome variable (thickness at FRC, thickening fraction, and dome excursion) remained unchanged. Based on these findings, we have not revised the content of the manuscript. Thank you for your valuable feedback.
Comment 7
Method
The accuracy and reliability of the ultrasound measurements are crucial for the present work. A detailed analysis of such procedures is demanded.
Response 7
Thank you for your feedback. The diaphragm ultrasound techniques used in this study were based on several previous studies. Since the methodology is described in detail in the Methods section, we have added the relevant references to support this.
Before Revision (Lines 109-110)
The diaphragm ultrasound examination was performed by a physician or an ultrasound technician with sufficient training.
After Revision (Lines 120-122)
The diaphragm ultrasound examination was performed by a physician or an ultrasound technician with sufficient training, following a protocol prepared with reference to previous studies [26-28].
Comment 8
Method
Replace height with stature and body weight with body mass.
Response 8
Thank you for your feedback. We have revised the terminology by replacing “height” with “stature” and “body weight” with “body mass.”
After Revision
stature : Line 32, 98, 112, 150, 157, Table1, 176, Table3, 208, Table5, 234, 239, 279, 306, 307, 310
body mass: 160, Table1
Comment 9
Method
The statistics are equivocal, as nothing is mentioned to support the use of parametric tests. In addition, the description of the multivariate analysis is poor. No caution was exercised to include some variables. For instance, the sarcopenia definition includes all your covariates, and a large colinearity may exist. Furthermore, repeating t-tests is not recommended as it increases the odds of errors. Effect sizes and confidence intervals are required. There is also a bias in your data since the sarcopenia group was older. Thus, it is not known whether such changes are age-related or not. In addition, mixing women and men may have also inserted a bias as the small number of women (in a reduced group; 3 times smaller) may have implications.
Response 9
・Regarding Variable Selection
Thank you for your feedback. The explanatory variables in the multivariate analysis were selected based on previous studies on sarcopenia and diaphragm ultrasound, as well as their clinical importance *1,2,3. As you correctly pointed out, handgrip strength and walking speed are part of the diagnostic criteria for sarcopenia. To avoid multicollinearity, these variables were not included in the analysis.
*1 Zeng B, He S, Lu H, Liang G, Ben X, Zhong W, et al. Prediction of Loss of Muscle Mass in Sarcopenia Using Ultrasonic Diaphragm Excursion. Contrast Media Mol Imaging. 2021;2021:4754705.
*2 Lee Y, Son S, Kim DK, Park MW. Association of Diaphragm Thickness and Respiratory Muscle Strength With Indices of Sarcopenia. Ann Rehabil Med. 2023;47(4):307-14.
*3 Deniz O, Coteli S, Karatoprak NB, Pence MC, Varan HD, Kizilarslanoglu MC, et al. Diaphragmatic muscle thickness in older people with and without sarcopenia. Aging Clin Exp Res. 2021;33(3):573-80.
・Regarding Parametric Tests
Thank you for your feedback. Although it was not mentioned in the main text, we confirmed the normality of the dependent variables—thickness at FRC, thickening fraction, and dome excursion—using histograms and Q-Q plots. Based on this confirmation, we determined that parametric tests were appropriate and conducted the analysis accordingly. Based on the above, we have not made any revisions to the content of this paper. Thank you for your valuable comments.
・Regarding Multicollinearity
Thank you for your feedback. I have consulted with the statistician who is a co-author of this study. We assessed multicollinearity using the Variance Inflation Factor (VIF). In all models, no variables showed a VIF exceeding 10, indicating that multicollinearity was not a significant issue. Therefore, we concluded that the selected explanatory variables were appropriate for the analysis.
・Regarding Repeated t-Tests
Thank you for your feedback. After consulting with the statistician who is a co-author of this study, we decided to revise Tables 2 and 4 to include group differences and confidence intervals instead. The p-values have been removed accordingly. We appreciate your valuable suggestions.
Thank you for your feedback. Please refer to Tables 2 and 4 for the revisions, where we have included group differences and confidence intervals while removing the p-values.
Before Revision (Lines 27-31)
Overall, 148 patients (mean age 78.1 years; sarcopenia, n=35; non-sarcopenia, n=103) were included. Thickness and TF did not differ significantly, but DE during DB was lower in the sarcopenia group (40.4 vs. 45.3 mm, p=0.09). The low SMI group had significantly lower thickness (1.91 vs. 2.13 mm, p=0.024) and DE (38.9 vs. 48.2 mm, p<0.001) than the normal SMI group.
After Revision (Lines 27-30)
Overall, 148 patients (mean age 78.1 years; sarcopenia, n=35; non-sarcopenia, n=103) were included. No statistically significant differences in diaphragm thickness, TF (thickness fraction), or DE were observed between the sarcopenia group and the non-sarcopenia group.
Before Revision (Lines 30-31)
The low SMI group had significantly lower thickness (1.91 vs. 2.13 mm, p=0.024) and DE (38.9 vs. 48.2 mm, p<0.001) than the normal SMI group.
After Revision (Lines 30-32)
The low SMI group had significantly lower thickness (difference -0.22, 95%CI ; -0.41, -0.29) and DE (difference -9.2, 95%CI ; -14.0, -4.49) than the normal SMI group.
Before Revision (Lines 152-156)
There were no statistically significant differences in diaphragm thickness (FRC) and TF between the sarcopenia group and the non-sarcopenia group (2.05 vs. 2.03 mm, p=0.85, and 95.4% vs. 87.2%, p=0.44, respectively). The DE during DB showed a trend toward be-ing statistically smaller in the sarcopenia group than in the non-sarcopenia group (40.4 vs. 45.3 mm, p=0.09) (Table 2).
After Revision (Lines 170-171)
There were no statistically significant differences in diaphragm thickness (FRC) , TF and DE during DB between the sarcopenia group and the non-sarcopenia group(Table 2).
Before Revision (Lines 157-161)
In the simple regression analysis of the relationship between the presence or absence of sarcopenia and thickness (FRC), TF, and DE during DB, no association was observed be-tween the presence of sarcopenia and thickness (FRC) (p=0.85) or TF (p=0.44). However, there was a trend toward a negative association between sarcopenia and DE during DB (p=0.09) (Table 3-1).
After Revision (Lines 172-175)
In the simple regression analysis of the relationship between the presence or absence of sarcopenia and thickness (FRC), TF, and DE during DB, no association was observed be-tween the presence of sarcopenia and thickness (FRC) (p=0.85) or TF (p=0.44) or DE during DB (p=0.09) (Table 3-1).
Before Revision (Lines 183-187)
Diaphragm thickness (FRC) was significantly lower in the low SMI group than in the normal SMI group (1.91 vs. 2.13 mm, p=0.024). TF showed no statistically significant difference between the low SMI group and the normal SMI group (95% vs. 85%, p=0.30). DE during DB was significantly smaller in the low SMI group than in the normal SMI group (38.9 vs. 48.2 mm, p<0.001) (Table 4).
After Revision (Lines 197-201)
Diaphragm thickness (FRC) was significantly lower in the low SMI group than in the normal SMI group (difference -0.22, 95%CI ; -0.41, -0.29). TF showed no statistically significant difference between the low SMI group and the normal SMI group (difference 9.8, 95%CI ; -8.8, 28.4). DE during DB was significantly smaller in the low SMI group than in the normal SMI group (difference -9.2, 95%CI ; -14.0, -4.49) (Table 4).
Before Revision (Lines 215-219)
This study investigated the associations of sarcopenia and low SMI with diaphragm thickness (FRC), TF, and DE during DB in elderly individuals aged ≥65 years. Compared with the non-sarcopenia group, the sarcopenia group showed a trend toward smaller DE during DB. Additionally, in the multivariable analysis adjusted for age, sex, and height, DE during DB was significantly negatively associated with sarcopenia.
After Revision (Lines 230-234)
This study investigated the associations of sarcopenia and low SMI with diaphragm thickness (FRC), TF, and DE during DB in elderly individuals aged ≥65 years. Compared with the non-sarcopenia group, the sarcopenia group showed no statistically significant differences. However, in the multivariable analysis adjusted for age, sex, and height, DE during DB was significantly negatively associated with sarcopenia.
・Regarding the Effects of Aging and Sex
Thank you for your feedback. As you correctly pointed out, the sarcopenia and non-sarcopenia groups in this study include older participants. Additionally, the influence of sex was also considered. Therefore, age and sex were included as adjustment variables in the multivariate analysis. Please refer to Tables 3 and 5 for details.
Comment 10
Result
The results are misinterpreted as non-significant differences are treated as “a trend”. This is inappropriate and misleading and unnaceptable.
Response 10
Thank you for your feedback. After consulting with the statistician who is a co-author of this study, we revised the phrase “a trend” to “no significant difference.”
Before Revision (Lines 154-156)
The DE during DB showed a trend toward being statistically smaller in the sarcopenia group than in the non-sarcopenia group (40.4 vs. 45.3 mm, p=0.09)
After Revision (Lines 170-171)
There were no statistically significant differences in diaphragm thickness (FRC) , TF and DE during DB between the sarcopenia group and the non-sarcopenia group.
Before Revision (Lines 157-161)
In the simple regression analysis of the relationship between the presence or absence of sarcopenia and thickness (FRC), TF, and DE during DB, no association was observed be-tween the presence of sarcopenia and thickness (FRC) (p=0.85) or TF (p=0.44). However, there was a trend toward a negative association between sarcopenia and DE during DB (p=0.09)
After Revision (Lines 172-175)
In the simple regression analysis of the relationship between the presence or absence of sarcopenia and thickness (FRC), TF, and DE during DB, no association was observed be-tween the presence of sarcopenia and thickness (FRC) (p=0.85) or TF (p=0.44) or DE during DB (p=0.09).
Before Revision (Lines 216-219)
Compared with the non-sarcopenia group, the sarcopenia group showed a trend toward smaller DE during DB. Additionally, in the multivariable analysis adjusted for age, sex, and height, DE during DB was significantly negatively associated with sarcopenia.
After Revision (Lines 231-234)
Compared with the non-sarcopenia group, the sarcopenia group showed no statistically significant differences. However, in the multivariable linear regression analysis adjusted for age, sex, and stature, DE during DB was significantly negatively associated with sarcopenia.
Comment 11
Result
In your multivariate analysis, colinearity must be reemphasized as an important issue. How was it controlled? In addition, table 3.2 does not present all the parameters necessary to characterize the analysis properly.
Response 11
Thank you for your feedback. I have consulted with the statistician who is a co-author of this study. We verified multicollinearity among the dependent variables, sarcopenia status, age, sex, and height.
・In the Thickness at FRC model, no variables showed a VIF exceeding 10 (height: 2.2, sex: 2.1, age: 1.23, sarcopenia status: 1.15).
・In the Thickening Fraction model, no variables showed a VIF exceeding 10 (height: 2.2, sex: 2.1, age: 1.23, sarcopenia status: 1.15).
・In the Dome Excursion model, no variables showed a VIF exceeding 10 (height: 2.17, sex: 2.08, age: 1.21, sarcopenia status: 1.14).
Based on these results, we concluded that the impact of multicollinearity was minimal. Therefore, we have not made any revisions to the content of the manuscript. Thank you for your valuable comments.
Comment 12
Result
There are two tables 3.
Response 12
Thank you for your feedback. The relevant section has been revised.
Reviewer 5 Report
Comments and Suggestions for Authors
The paper is generally well-written and structured. However, in my opinion, there are several issues to reconsider.
I suggest the authors include the study type in the title.
Introduction: Please provide the details regarding the DE e.g., how DE relates to diaphragm function or respiration.
Method: Please provide the details regarding how to perform DB. Are there differences from maximum inspiration? Also, how did you know those participants can perform the total lung capacity? Please provide the instructions while performing the diaphragm ultrasound.
Discussion: In this study, there were no statistically significant differences in diaphragm thickness (FRC) and TF, even though %VC is higher in non-sarcopenia compared to sarcopenia groups. Please explain.
It found that height had no significant differences between no-sarcopenia and sarcopenia. However, in Table 3 and Table 5, DE was associated with height but not in diaphragm thickness at FRC and thickening fraction. Can you explain these results?
Line 263-267: you stated that "A previous study involving healthy older adults indicated that TF was not affected by factors such as sex, height, and BMI [31]. The results of the present study also indicated that TF was not affected by sarcopenia or reduced skeletal muscle mass. Therefore, TF could potentially be used as a consistent reference across a wide range of patients, and it could be considered in the diagnostic criteria for diaphragmatic paralysis [23]." In lines 265-267, it makes me confused.
If TF was not related to sarcopenia or skeletal muscle mass, why did you conclude that TF could be considered in the diagnostic criteria?
Please provide the biomechanics of why DE-related to diaphram or sarcopenia.
Author Response
Comment 1
Title
I suggest the authors include the study type in the title.
Response 1
Thank you for your feedback. We have added “Cross-Sectional Study” to the end of the title.
Before Revision (Lines 2-3)
Relationship Between Diaphragm Function and Sarcopenia Assessed by Ultrasound ; An Ultrasound study
After Revision (Lines 2-3)
Relationship Between Diaphragm Function and Sarcopenia Assessed by Ultrasound ; A Cross sectional study
Comment 2
Introduction
Introduction: Please provide the details regarding the DE e.g., how DE relates to diaphragm function or respiration.
Response 2
Thank you for your feedback. As you pointed out, there have been several reports on the relationship between diaphragm ultrasound indices and respiratory muscles *1,2. To improve clarity, we have added this information to the Introduction.
*1 Summerhill EM, Angov N, Garber C, McCool FD. Respiratory muscle strength in the physically active elderly. Lung. 2007;185(6):315-20.
*2 Yamada T, Minami T, Yoshino S, Emoto K, Mabuchi S, Hanazawa R, et al. Relationship Between Diaphragm Thickness, Thickening Fraction, Dome Excursion, and Respiratory Pressures in Healthy Subjects: An Ultrasound Study. Lung. 2024;202(2):171-8.
Before Revision (Lines 56-59)
One involves measuring the diaphragm thickness (DT) and the change in thickness due to respiration, known as the thickening fraction (TF). The other measures the movement of the dome of the diaphragm during respiration, known as diaphragm excursion (DE) [18].
After Revision (Lines 57-65)
One involves measuring the diaphragm thickness (DT) and the change in thickness due to respiration, known as the thickening fraction (TF). The other measures the movement of the dome of the diaphragm during respiration, known as diaphragm excursion (DE) [18] . Previous studies have reported a positive association be-tween diaphragm thickness (DT) and maximal inspiratory pressure (MIP) [19], as well as a correlation between diaphragm excursion (DE) and both MIP and maximal expiratory pressure (MEP) [20]. These findings suggest the potential for diaphragm ultrasound parameters to assess respiratory muscle strength and respiratory sarcopenia. 
Comment 3
Method
・Please provide the details regarding how to perform DB. Are there differences from maximum inspiration?
・Also, how did you know those participants can perform the total lung capacity?
・Please provide the instructions while performing the diaphragm ultrasound.
Response 3
Thank you for your feedback. For diaphragmatic breathing (DB), the instructions given were: “Take a deep breath in, and once you reach full inhalation, exhale and relax.” The same instructions were applied for maximal inspiration. This information has been added to the manuscript.
Participants who were unable to follow these instructions were excluded, and we interpreted maximal inspiration as representing TLC (total lung capacity). Previous studies on the consensus for diaphragm ultrasound have also emphasized the need to standardize methods to elicit patients' maximal effort*. Further research is needed to establish standardized instructions in the future.
* Haaksma ME, Smit JM, Boussuges A, et al. EXpert consensus On Diaphragm UltraSonography in the critically ill (EXODUS): a Delphi consensus statement on the measurement of diaphragm ultrasound-derived parameters in a critical care setting. Crit Care. 2022;26(1):99.
Before Revision (Lines 114-116)
The DT was measured at the end of quiet expiration (functional residual capacity [FRC]) and at maximum inspiration (total lung capacity [TLC]).
After Revision (Lines 126-130)
The DT was measured at the end of quiet expiration (functional residual capacity [FRC]) and at maximum inspiration (total lung capacity [TLC]). During DT measurement, verbal instructions were given as follows: "Take a deep breath in, and at the end of the inhalation, exhale and relax." Deep inspiration was defined as TLC, and the end of exhalation was defined as FRC.
Before Revision (Lines 124-126)
The M-mode line was adjusted to be as perpendicular as possible to the diaphragm dome, and the difference in movement of the dome during deep breathing (DB) was measured [23].
After Revision (Lines 138-142)
The M-mode line was adjusted to be as perpendicular as possible to the diaphragm dome, and the difference in movement of the dome during deep breathing (DB) was measured [28] . The verbal in-structions for measuring DE during DB were the same as those for DT measurement: "Take a deep breath in, and at the end of the inhalation, exhale and relax."
Comment 4
Discussion
In this study, there were no statistically significant differences in diaphragm thickness (FRC) and TF, even though %VC is higher in non-sarcopenia compared to sarcopenia groups. Please explain.
Response 4
Thank you for your feedback. It has been reported that sarcopenia is associated with VC *. However, the relationship between %VC and thickness at FRC or thickening fraction (TF) remains unclear, and thus %VC was not included as an explanatory variable in the multivariate analysis.
As a precaution, we conducted the analysis with %VC included, but no statistically significant associations were observed with thickness at FRC (p = 0.575) or TF (p = 0.065). Based on these results, we have not made any changes to the content of the manuscript. Thank you for your valuable input.
* Kera T, Kawai H, Ejiri M, Ito K, Hirano H, Fujiwara Y, et al. Comparison of Characteristics of Definition Criteria for Respiratory Sarcopenia-The Otassya Study. Int J Environ Res Public Health. 2022;19(14).
Comment 5
It found that height had no significant differences between no-sarcopenia and sarcopenia. However, in Table 3 and Table 5, DE was associated with height but not in diaphragm thickness at FRC and thickening fraction. Can you explain these results?
Response 5
Thank you for your feedback. As you pointed out, while height was not associated with thickness at FRC or thickening fraction (TF), it was associated with dome excursion (DE). The precise reason for this relationship remains unclear. Previous studies have reported that height does not influence DE in younger individuals *, suggesting that this association may be unique to older adults.
One possible explanation is that vertebral deformities, such as kyphosis due to decreased bone density, may affect height and subsequently influence DE. However, this remains a hypothesis and requires further investigation. Given the importance of this observation, we have added this point to the Discussion section. Thank you for your valuable input.
* Yamada T, Minami T, Yoshino S, Emoto K, Mabuchi S, Hanazawa R, et al. Relationship Between Diaphragm Thickness, Thickening Fraction, Dome Excursion, and Respiratory Pressures in Healthy Subjects: An Ultrasound Study. Lung. 2024;202(2):171-8.
Before Revision (Lines 284-286)
This may reflect the age-related reduction in fast twitch fibres and could manifest as a decrease in DE during DB.
After Revision (Lines 304-311)
This may reflect the age-related reduction in fast twitch fibres and could manifest as a de-crease in DE during DB. In addition, multivariable linear regression analysis revealed that DE was statistically significantly associated with stature. Previous studies have re-ported that stature does not influence DE in younger individuals [20], suggesting that this association may be specific to older adults.One possible factor is vertebral deformity, such as kyphosis, caused by decreased bone density, which may affect stature and consequently influence DE. However, vertebral deformities were not measured in this study, and further investigations are necessary to clarify this relationship.
Comment 6
Line 263-267: you stated that "A previous study involving healthy older adults indicated that TF was not affected by factors such as sex, height, and BMI [31]. The results of the present study also indicated that TF was not affected by sarcopenia or reduced skeletal muscle mass. Therefore, TF could potentially be used as a consistent reference across a wide range of patients, and it could be considered in the diagnostic criteria for diaphragmatic paralysis [23]." In lines 265-267, it makes me confused.If TF was not related to sarcopenia or skeletal muscle mass, why did you conclude that TF could be considered in the diagnostic criteria?
Response 6
Thank you for your feedback. Regarding the statement, “The results of the present study also indicated that TF was not affected by sarcopenia or reduced skeletal muscle mass. Therefore, TF could potentially be used as a consistent reference across a wide range of patients, and it could be considered in the diagnostic criteria for diaphragmatic paralysis [23],” it was intended to refer to the diagnostic criteria for diaphragmatic paralysis, not sarcopenia. To improve clarity, we have revised it as follows.
Before Revision (Lines 264-267)
The results of the present study also indicated that TF was not affected by sarcopenia or reduced skeletal muscle mass. Therefore, TF could potentially be used as a consistent reference across a wide range of patients, and it could be considered in the diagnostic criteria for diaphragmatic paralysis [23].
After Revision (Lines 280-284)
The results of the present study also indicated that TF was not affected by sarcopenia or reduced skeletal muscle mass. TF may not be suitable for the diagnosis of sarcopenia or reduced skeletal muscle mass; however, as it is less influenced by factors such as age, height, and sex, it has the potential to be applicable as a diagnostic criterion for diaphragmatic paralysis across a wide range of patients [28].
Comment 7
Please provide the biomechanics of why DE-related to diaphragm or sarcopenia.
Response 7
Thank you for your feedback. The diaphragm is a primary inspiratory muscle, and reduced diaphragmatic function is known to result in a decrease in maximal inspiratory pressure (MIP). Previous studies have reported an association between dome excursion (DE) during diaphragmatic breathing (DB) and MIP. The proposed mechanism is that greater DE leads to a larger change in thoracic volume, which increases the negative pressure applied to the lungs, thereby enhancing MIP *. This point has been added to the Discussion section for clarification.
* Yamada T, Minami T, Yoshino S, Emoto K, Mabuchi S, Hanazawa R, et al. Relationship Between Diaphragm Thickness, Thickening Fraction, Dome Excursion, and Respiratory Pressures in Healthy Subjects: An Ultrasound Study. Lung. 2024;202(2):171-8.
Before Revision (Lines 276-277)
In humans, a positive correlation between maximum inspiratory pressure and DE during DB has also been reported [32].
After Revision (Lines 293-297)
In humans, a positive correlation between maximum inspiratory pressure and DE during DB has also been reported. The mechanism is thought to involve a larger DE resulting in a greater change in thoracic cavity volume, which increases the negative pressure exerted on the lungs, ultimately leading to an increase in MIP [20].
Round 2
Reviewer 2 Report
Comments and Suggestions for Authors
I appreciate the authors' efforts to provide responses to the comments.
Author Response
Dear reviewer
Thank you for your response and for providing valuable comments on my manuscript. Your insights have allowed me to conduct a deeper analysis of the paper. Thank you very much for taking the time to review my manuscript despite your busy schedule. I sincerely appreciate your support and look forward to your continued guidance.
Reviewer 3 Report
Comments and Suggestions for Authors
The Authors modified the test as request by the referee and the study improved.
Author Response

(The authors gave the same response as above.)

Reviewer 4 Report
Comments and Suggestions for Authors
I thank the authors for their responses. Let me help clarify some points and rebate some arguments that still require amendments.
In the previous round, I mentioned that DT, TF, and DE are not muscle function indicators but morphological parameters and required the authors to adjust the study’s rationale. The authors replied that “a positive association between diaphragm thickness (DT) and maximal inspiratory pressure (MIP)” has been found [Reference 19]. A visit to that paper reveals the following: “There was a positive association between PImax and tdi (r = 0.43, p = 0.03).” Such assertiveness requires caution, as a positive correlation of 0.42 explains around 17% of your phenomena, irrespective of its significance. Therefore, I still have issues with the association or correlation the authors and others advocate. The same applies when associating the other morphological and functional (respiratory) variables. Your reference # 20 supports the same idea I am pointing to.
In the second point, I addressed the rationale of measuring parameters on the same sample and indicated that the connection of these parameters with sarcopenia should be emphasized. Establishing the connection between your key variables and sarcopenia is still missing. Again, after reading Deniz’s study, I am still confused regarding the differences between the current study and what they have done. Is it related only to TF? If so, please, explain and clear such doubt.
In the third point, I asked about the sample size calculations. However, you did not make such a mandatory request in the manuscript, which does not solve the question. You may add such calculations using a posteriori approach.
I thank the authors for including some of the indicated references. A more critical approach to using HGS as a parameter may be included in the limitations section of the study. Also, based on such arguments, exercise some caution while interpreting sarcopenia and how it relates to your variables and outcomes.
I know the literature and how to find reliability measurements. However, I am asking about the reliability of your study, not others. Can you guarantee that the experimenter in your study performed as accurately as others? What is the variance of your examiner? Note that such information is relevant to ensure the reader that your measurements are reliable, irrespective of his/her experience.
You better seek professional consultancy regarding stats. I haven’t requested to remove the p values. They had to be adjusted (corrected) according to several strategies (e.g., Bonferroni).
Author Response
Dear reviewer
Thank you very much for taking the time to read my manuscript and for providing valuable comments. I have prepared responses to the comments below, and I would greatly appreciate it if you could kindly review them.
Comment 1
In the previous round, I mentioned that DT, TF, and DE are not muscle function indicators but morphological parameters and required the authors to adjust the study’s rationale. The authors replied that “a positive association between diaphragm thickness (DT) and maximal inspiratory pressure (MIP)” has been found [Reference 19]. A visit to that paper reveals the following: “There was a positive association between PImax and tdi (r = 0.43, p = 0.03).” Such assertiveness requires caution, as a positive correlation of 0.42 explains around 17% of your phenomena, irrespective of its significance. Therefore, I still have issues with the association or correlation the authors and others advocate. The same applies when associating the other morphological and functional (respiratory) variables. Your reference # 20 supports the same idea I am pointing to.
*19  Summerhill EM, Angov N, Garber C, McCool FD. Respiratory muscle strength in the physically active elderly. Lung. 2007;185(6):315-20.
*20  Yamada T, Minami T, Yoshino S, Emoto K, Mabuchi S, Hanazawa R, et al. Relationship Between Diaphragm Thickness, Thickening Fraction, Dome Excursion, and Respiratory Pressures in Healthy Subjects: An Ultrasound Study. Lung. 2024;202(2):171-8.
Response 1
Thank you for your valuable feedback. As you pointed out, the correlation coefficient in *19 is not high, and it may not be appropriate to assert a positive correlation. Therefore, we have made some modifications to the wording. Regarding the association with DE, it has been reported as a statistically significant difference, so we did not make any changes to the description. Thank you for your valuable feedback.
Before Revision (Lines 57-65)
One involves measuring the diaphragm thickness (DT) and the change in thickness due to respiration, known as the thickening fraction (TF). The other measures the movement of the dome of the diaphragm during respiration, known as diaphragm excursion (DE) [18]. Previous studies have reported a positive association be-tween diaphragm thickness (DT) and maximal inspiratory pressure (MIP) [19], as well as a correlation between diaphragm excursion (DE) and both MIP and maximal expiratory pressure (MEP) [20]. These findings suggest the potential for diaphragm ultrasound parameters to assess respiratory muscle strength and respiratory sarcopenia. 
After Revision (Lines 57-65)
One involves measuring the diaphragm thickness (DT) and the change in thickness due to respiration, known as the thickening fraction (TF). The other measures the movement of the dome of the diaphragm during respiration, known as diaphragm excursion (DE) [18]. Previous studies have not clearly established the association between diaphragm thickness (DT) and maximal inspiratory pressure (MIP) [19, 20], but a correlation between diaphragm excursion (DE) and both MIP and maximal expiratory pressure (MEP) has been reported [20]. These findings suggest the potential for diaphragm ultrasound parameters to assess respiratory muscle strength and respiratory sarcopenia. 
Comment 2
In the second point, I addressed the rationale of measuring parameters on the same sample and indicated that the connection of these parameters with sarcopenia should be emphasized. Establishing the connection between your key variables and sarcopenia is still missing. Again, after reading Deniz’s study, I am still confused regarding the differences between the current study and what they have done. Is it related only to TF? If so, please, explain and clear such doubt.
Response 2
Thank you for your feedback. As you mentioned, Deniz et al. reported on the association with DT based on the EWGSOP international criteria. However, no studies have validated DE and TF based on the international criteria for sarcopenia. This study evaluates TF, DE, and DT of diaphragmatic ultrasound indices in accordance with the international criteria established by AWGS, using the same cohort of participants. We recognize the novelty of our study in the simultaneous measurement of three parameters, including TF and DE, based on international criteria, and the evaluation of their interrelationships. We would greatly appreciate your understanding of this perspective.
Comment 3
I asked about the sample size calculations. However, you did not make such a mandatory request in the manuscript, which does not solve the question. You may add such calculations using a posteriori approach.
Response 3
Thank you for your valuable feedback. We have consulted with the statistical expert involved in this study. As this is a cross-sectional study aimed at hypothesis generation, we conducted the analysis based on the data collected during the study period. In this context, we determined that a priori sample size calculation was not essential. Additionally, regarding post hoc sample size calculation, it has been noted in several publications that such an approach is not recommended.*1-3 Therefore, we concluded that it would be preferable not to include this in the manuscript. We hope for your kind understanding.
*1 Althouse AD. Post Hoc Power: Not Empowering, Just Misleading. J Surg Res. 2021;259:A3-a6.
*2 Heinsberg LW, Weeks DE. Post hoc power is not informative. Genet Epidemiol. 2022;46(7):390-4.
*3 John M Hoenig, Dennis M Heisy. The Abuse of Power. The Amrican Statistician. 2012 ; 19-24
Comment 4
I thank the authors for including some of the indicated references. A more critical approach to using HGS as a parameter may be included in the limitations section of the study. Also, based on such arguments, exercise some caution while interpreting sarcopenia and how it relates to your variables and outcomes.
Response 4
Thank you for your feedback. I have discussed this matter with the co-authors. As a result of our discussions, we concluded that a more detailed examination of grip strength falls outside the primary focus of this study. Therefore, We have limited our discussion to brief comments. We appreciate your understanding.
Comment 5
I know the literature and how to find reliability measurements. However, I am asking about the reliability of your study, not others. Can you guarantee that the experimenter in your study performed as accurately as others? What is the variance of your examiner? Note that such information is relevant to ensure the reader that your measurements are reliable, irrespective of his/her experience.
Response 5
Thank you for your feedback. In this study, inter-observer variability was not measured. Therefore, variations between examiners could not be verified. This is an important point, and I have included it in the Limitations section. Thank you for your valuable feedback.
Before Revision (Lines 337-341)
Structural changes in diaphragm fibres have been reported with aging [42], and skeletal muscle loss in younger individuals may yield different results, warranting further investigation. Second, the study included patients who visited medical institutions, and we did not examine the possibility that underlying conditions may have influenced the results.
After Revision (Lines 337-344)
Structural changes in diaphragm fibres have been reported with aging [42], and skeletal muscle loss in younger individuals may yield different results, warranting further investigation.
Second, although the ultrasound examinations were conducted by pre-trained examiners, inter-observer variability was not evaluated, and differences in measurement accuracy between examiners have not been verified. Third, the study included patients who visited medical institutions, and we did not examine the possibility that underlying conditions may have influenced the results.
Comment 6
You better seek professional consultancy regarding stats. I haven’t requested to remove the p values. They had to be adjusted (corrected) according to several strategies (e.g., Bonferroni).
Response 6
Thank you for your valuable feedback. We have consulted with the statistical expert involved in this study. This study is primarily exploratory in nature, aiming not to test hypotheses based on p-values, but rather to generate new hypotheses and identify trends from the data. Moreover, adjusting for multiplicity and applying stricter significance levels can reduce statistical power, potentially leading to missed detection of true effects. For these reasons, we had determined that it was not necessary to adjust for multiplicity in this study. We hope this approach is acceptable and would appreciate your understanding.
Reviewer 5 Report
Comments and Suggestions for Authors
The revised manuscript can be accepted.
Author Response

(The authors gave the same response as above.)
